# Exploring the Design Space of Transition Matching

**Uriel Singer, Yaron Lipman**
FAIR at Meta

## Abstract

Transition Matching (TM) is an emerging paradigm for generative modeling that generalizes diffusion and flow-matching models as well as continuous-state autoregressive models. TM, similar to previous paradigms, gradually transforms noise samples to data samples, however it uses a second "internal" generative model to implement the transition steps, making the transitions more expressive compared to diffusion and flow models. To make this paradigm tractable, TM employs a large backbone network and a smaller "head" module to efficiently execute the generative transition step. In this work, we present a large-scale, systematic investigation into the design, training and sampling of the head in TM frameworks, focusing on its time-continuous bidirectional variant. Through comprehensive ablations and experimentation involving training 56 different 1.7B text-to-image models (resulting in 549 unique evaluations) we evaluate the affect of the head module architecture and modeling during training as-well as a useful family of stochastic TM samplers. We analyze the impact on generation quality, training, and inference efficiency. We find that TM with an MLP head, trained with a particular time weighting and sampled with high frequency sampler provides best ranking across all metrics reaching state-of-the-art among all tested baselines, while Transformer head with sequence scaling and low frequency sampling is a runner up excelling at image aesthetics. Lastly, we believe the experiments presented highlight the design aspects that are likely to provide most quality and efficiency gains, while at the same time indicate what design choices are not likely to provide further gains.

## 1 Introduction

Transition Matching (TM) Shaul et al. (2025) is a recent generalization of several media generative paradigms including diffusion models Sohl-Dickstein et al. (2015); Ho et al. (2020); Song et al. (2020), flow matching models Lipman et al. (2022); Liu et al. (2022); Albergo & Vanden-Eijnden (2022), and continuous-state autoregressive image generation Li et al. (2024); Team et al. (2025) that offers new design choices that go beyond the scope of these former paradigms and already shown to yield improved image quality and/or more efficient sampling at inference time.

In this work we focus on TM's continuous time bidirectional variant, which, similarly to previous paradigms, learns a transition function (kernel) that gradually transfers a source (noise) sample $X_0$ to a target (data) sample $X_1$ by iteratively producing future samples $X_{t'}$ from previous samples $X_t$, $0 \le t < t' \le 1$. Differently from previous work, TM models the transition kernel with a second "internal" generative model, offering a more expressive transition kernels than, e.g., diffusion models that utilize a factorized (i.e., independent in each coordinate) multivariate Gaussian as kernels. To keep things tractable, TM adopts a backbone–head paradigm, in which: The *backbone* (typically a large transformer) encodes current state $X_t$ as well as conditioning information, producing a rich latent representation per input token. The *head* (typically much smaller than the backbone) serves as a learnable module tasked with translating backbone latent representations into concrete transition outputs, producing the next state $X_{t'}$ with $t' > t$. While backbone architecture for diffusion models have been, and still are, thoroughly investigated (e.g., Peebles & Xie (2022)), systematic exploration of head architecture and hyperparameters is lacking in the current literature. Most existing works treat the head as a fixed, minimal component—often a single MLP or a lightweight mapping—without investigating how variations in design might impact model behavior and efficiency (Li et al., 2024; Fan et al.; Team et al., 2025; Shaul et al., 2025). In fact, due to its particular role in the generative

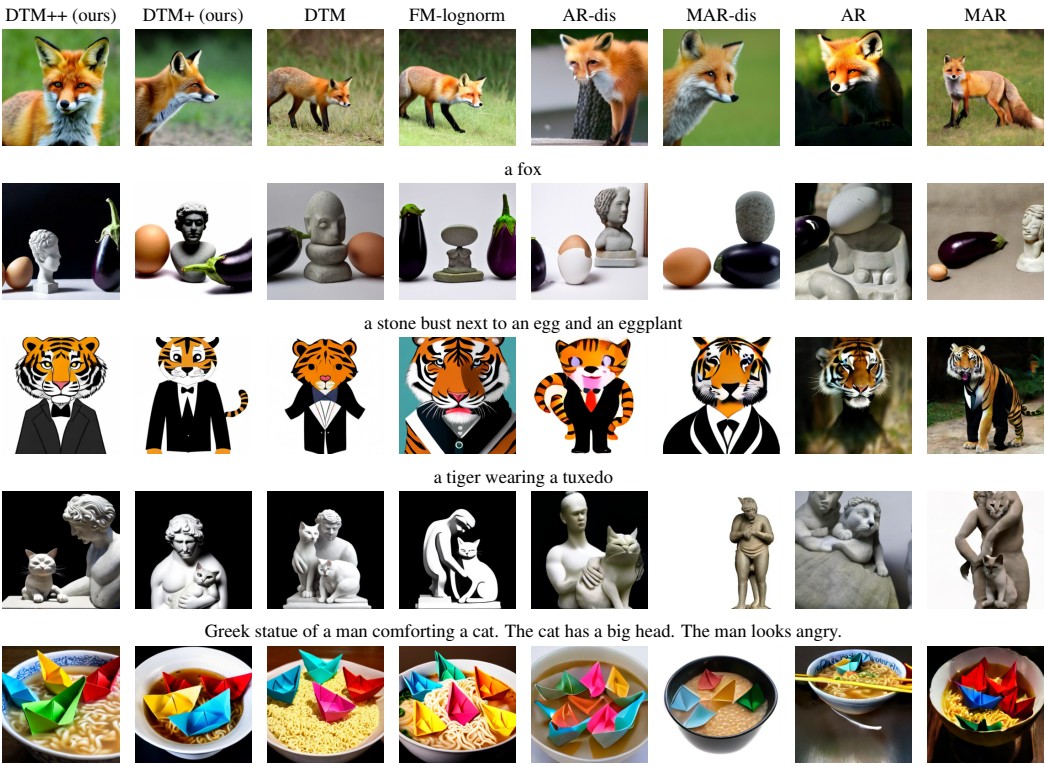

| DTM++ (ours) | DTM+ (ours) | DTM | FM-lognorm | AR-dis | MAR-dis | AR | MAR |

a fox

a stone bust next to an egg and an eggplant

a tiger wearing a tuxedo

Greek statue of a man comforting a cat. The cat has a big head. The man looks angry.

A high resolution photo of a large bowl of ramen. There are several origami boats in the ramen of different colors.

Figure 1: Samples comparing of our Best D-TM MLP (DTM++) and Transformer (DTM+) Models with DTM baseline, FM-lognormal, AR, MAR, AR-discrete, MAR-discrete as baselines. All models share similar architecture and training recipe.

process and its small relative size, the head design holds much potential for improving the model performance by exploring head-specific architectures and different scaling laws. In this paper we take on this opportunity and explore different design choices for the head both in training and inference stages, with the goal of improving one or more of the three main properties of a highly performant generative model: generation quality, training efficiency, and inference efficiency. We offer the following contributions:

(i) *Comprehensive exploration:* We perform a large-scale (i.e., 56 different 1.7B unique model trains resulting in 549 unique evaluations), systematic ablation study of TM design space including exploration of head model architecture and size, sequence scaling laws, batch size scaling laws, time weighting, model parameterizations and inference algorithms. For fair comparison across models and baselines we keep backbone, training dataset, and most training hyperparameters strictly fixed, while evaluating all models on a comprehensive setup of 4 datasets with 25 individual metrics summed up to a single performance *rank*.

(ii) *TM sampling:* We design a novel stochastic sampling algorithm for Transition Matching that is shown to considerably improve generative quality while keeping the computational cost the same.

(iii) *Actionable Guidelines:* Our ablations illuminate tradeoffs and best practices for continuous-time bidirectional TM-based generative models. In a nutshell: TM with MLP head trained with particular backbone-head time weighting and sampled with high frequency stochastic sampler leads to the best ranking model (DTM++), where Transformer head with sequence scaling and low frequency sampling is the runner-up (DTM+) that excels in image aesthetics, see fig. 1.

## 2 BACKGROUND: CONTINUOUS-TIME TRANSITION MATCHING

**Goal and motivation of Transition Matching**   In this paper we focus on continuous-time fully-bidirectional variant of Transition Matching (TM) (Shaul et al., 2025), which we found to lead to best results in our text-to-image experimental setup. An image is encoded as a sequence of $n$ continuous tokens in dimension $d$, that is $x = (x^1, \ldots, x^n) \in \mathbb{R}^{n \times d}$. Capital letters are used to denote *random variables*. Similar to diffusion and flow models, TM learns a transition kernel,

$$X_{t'} \sim p_{t'|t}^\theta(\cdot \mid X_t), \quad 0 \le t < t' \le 1 \tag{1}$$

gradually transforming a source (e.g., noise) sample $X_0 \sim p_0$ to a target (e.g., data) sample $X_1 \sim p_1$, where $\theta$ denotes the learnable parameters. In diffusion models[1], transitions $p_{t'|t}^\theta$ have the simplified form of a factorized Gaussian kernel,

$$p_{t'|t}^\theta(\cdot \mid x) = \mathcal{N}(\cdot \mid \mu_t(x), \sigma_t^2 I), \tag{2}$$

where in flow matching this kernel is even simpler, i.e., a delta function. Transition Matching (TM) is a generalization of diffusion and flow models that utilizes *more expressive* transition kernels $p_{t'|t}^\theta$ which are modeled by an "internal" generative flow model that learns to sample $X_{t'}$ given state $X_t$. TM learns $p_{t'|t}^\theta$ by matching it to the transitions of some user defined supervision process denoted $q_{t'|t}$ defined next.

**Supervising process and kernel parameterization**   The TM model $p_{t'|t}^\theta$ is learned by regressing a *supervision process* $q$. A supervision process is any random process $(X_t)_{t \in [0,1]}$, with probability density $q$ such that its marginals, denoted by $q_t(x_t)$, at time $t = 0$ and $t = 1$ coincides with the desired source distribution $p_0$ and target distributions $p_1$, respectively. That is, $q_0 = p_0$ and $q_1 = p_1$. The choice of a supervision process $q$ is a design freedom of TM, where in this paper we follow the standard choice of the linear (a.k.a. conditional optimal transport) path (Lipman et al., 2022; Liu et al., 2022; Shaul et al., 2025):

$$X_t = (1 - t) X_0 + t X_1 \qquad \text{\textit{Linear supervising process}} \tag{3}$$

where $t \in [0, 1]$, $X_0 \sim \mathcal{N}(0, I)$ is a noise sample, and $X_1 \sim p_1$ is a data sample.

The conditional distribution $q_{t'|t}(x_{t'}|x_t)$ is the main object we want to regress in TM. That is, given a current state $X_t$ learn to sample $X_{t'}$ so it is distributed according to $q_{t'|t}(\cdot|X_t)$. However, it is often useful to learn to predict a different random quantity $Y$ given $X_t$ instead of directly $X_{t'}$. The reason is two-fold: First, avoid dealing with the extra time variable $t'$ during training; and second, improve inductive bias and performance by e.g., removing the dependence on $t$ in the target quantity. To make the $Y$ parameterization useful in practice, one needs to make sure that predicting $X_{t'}$ given samples $Y$ and $X_t$ is rather easy and computationally cheap. The parameterization process is justified mathematically by the law of total (conditional) probability,

$$q_{t'|t}(x_{t'}|x_t) = \int q_{t'|t,Y}(x_{t'}|x_t, y) p_{Y|t}(y|x_t) \mathrm{d}y. \tag{4}$$

That is, TM learns to sample $Y \sim p_{Y|t}(\cdot|x_t)$. After a sample of $Y$ is produced, the next state is sampled according to

$$X_{t'} \sim q_{t'|t,Y}(\cdot|X_t, Y), \tag{5}$$

which is guaranteed to have the distribution $q_{t'|t}(\cdot|X_t)$ by the law in eq. (4). The choice of $Y$ is another degree of freedom of TM. Shaul et al. (2025) made the choice of noise-data difference,

$$Y = X_1 - X_0 \qquad \text{\textit{D-TM}} \tag{6}$$

motivated by the relation

$$X_{t'} = X_t + (t' - t)(X_1 - X_0) \tag{7}$$

that holds for the linear process (eq. (3)); this variant is called Difference Transition Matching (D-TM). An equivalent parameterization is also explored in Zhang et al. (2025).

---

[1]We use the forward-time convention similar to flow matching, while standard diffusion models use backward-time convention moving from state $X_t$ to state $X_{t-1}$.

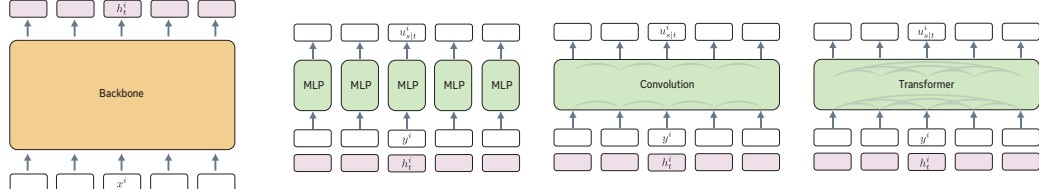

Figure 2: Head architectures explored (in green): MLP, Convolution, and Transformer. The backbone model (in orange) is kept fixed.

**Modeling** The goal in TM is to learn a model to sample $Y \sim p_{Y|t}(\cdot|X_t)$ for each $t \in [0, 1)$ and $X_t \sim q_t$. This entails an "interior" model that is resampled for each $t$. To make this tractable (and in fact, improve performance), TM uses a backbone-head architecture. That is, given a current state $X_t$ a *backbone* network computes a latent vector representation,

$$h_t = f_t^\varphi(X_t). \tag{8}$$

Next, this latent vector is used by a head network $u_{s|t}^\phi(\cdot|x_t)$ to sample $Y \sim p_{Y|t}(\cdot|X_t)$. The head will be used to sample $Y$ via flow matching (Lipman et al., 2022; Liu et al., 2022; Albergo & Vanden-Eijnden, 2022). That is, it is learned by minimizing the flow matching loss

$$\mathcal{L}_{\text{TM}}(\theta) = \mathbb{E}_{t, X_0, X_1, s, Y_0, Y_1} \left\| u_s^\phi(Y_s|h_t) - (Y_1 - Y_0) \right\|^2, \tag{9}$$

where $t, s \sim U(0, 1)$ uniformly and independently, $X_t = (1-t)X_0 + tX_1 \sim q_t$, $Y_0 \sim \mathcal{N}(0, I)$, $Y_1 \sim p_{Y|t}(\cdot|X_t)$, and $Y_s = (1-s)Y_0 + sY_1$. Once training is completed, sampling $Y \sim p_{Y|t}(\cdot|x_t)$ is done by solving the ordinary differential equation (ODE),

$$\frac{\mathrm{d}}{\mathrm{d}s}Y_t = u_{s|t}^\phi(Y_s|h_t) \tag{10}$$

starting with $Y_0 \sim \mathcal{N}(0, I)$ and solving until $s = 1$, with $h_t = f_t^\varphi(x_t)$. The total learnable parameters of the TM model are $\theta = (\varphi, \phi)$; for brevity, we sometimes omit the parameters superscript. The sampling pseudocode is provided in algorithm 2.

## 3 EXPLORING THE DESIGN SPACE

The main goal of this paper is to explore the design space of continuous-time Transition Matching for maximizing performance (i.e., quality and text adherence of generated images) and efficiency (i.e., inference and training speed). First we consider the training phase, focusing on head modeling and architecture. The head $u_{s|t}$ is responsible for sampling $Y$ given the current state $X_t$ encoded via a latent $h_t$ computed by the backbone $f_t$. The head introduces a useful leverage point as it can be chosen (as we will see) to be significantly smaller and faster than the backbone model and therefore exhibits scaling laws that can improve the overall performance and efficiency without a significant increase to the overall computational and memory costs. Second, we explore different inference options including efficiency-quality tradeoffs and a novel TM stochastic sampling algorithm. We start by describing our experimental setup (text-to-image generation), and then move to discuss the different design choices explored and the relevant experiments.

### 3.1 EXPERIMENTAL SETUP

We start with explaining the experimental setup that is fixed throughout the experiments.

**Backbone model and Data** We fix our backbone model $f_t^\varphi$ to a DiT transformer model (Peebles & Xie, 2022) with 24 layers (including self and cross attention) and a hidden dimension of $d_b = 2048$; this gives total of 1.7B parameters for the backbone. Our data consists of 350M text-image pairs. Each image is of dimension $256 \times 256 \times 3$; we move it to a latent representation as follows: we first embed it into a latent space using SDXL-VAE (Podell et al., 2023) to dimension $32 \times 32 \times 4$ and then $2 \times 2$ patched to get a latent representation of $16 \times 16 \times 16$ and therefore our data and noisy vectors $X, Y \in \mathbb{R}^{n \times d}$ where $n = k^2$ with $k = 16$ and $d = 16$. The standard hyperparameters are following Shaul et al. (2025) including optimizer, number of training iterations (500k), where the only difference is that we use learning rate decay. We use standard Classifier Free Guidance (CFG) (Ho & Salimans, 2022) training and sampling for flow matching, see appendix F.

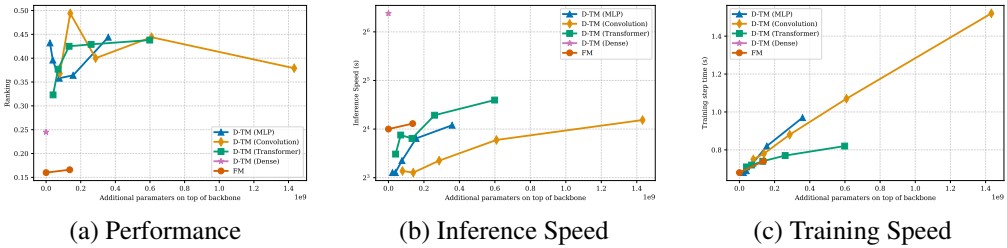

(a) Performance        (b) Inference Speed        (c) Training Speed

Figure 3: (a) Performance as a function of additional parameters. Increasing the head size does not improve performance. (b) Inference speed (seconds). (c) Training speed (iteration, seconds).

**Metrics** We report text–to-image metrics over four different prompt datasets: MS-COCO (Lin et al., 2014), PartiPrompts (Yu et al., 2022), GenEval (Ghosh et al., 2023), and T2ICompBench (Huang et al., 2023). Over MS-COCO and PartiPrompts we calculate standard image quality metrics: CLIPScore (Hessel et al., 2021), PickScore (Kirstain et al., 2023), ImageReward (Xu et al., 2023a), UnifiedReward (Wang et al., 2025), Aesthetic (aes, 2022), and DeQA (You et al., 2025). On GenEval and T2ICompBench we calculate their internal metrics and the corresponding overall scores. As there is a large amount of datasets (4) and metrics (25 different metrics), we aggregate all metrics into a single *rank* calculated for each evaluated model as follows. We rank each model to be tested (we have a total of 549 such models from 56 unique trains) according to each metric (scores from 1 to 549, where higher is better). We then average these ranks across all metrics and divide by the total number of models to get a final rank score in $[0, 1]$ for each model.

## 3.2 TRAINING AND HEAD MODELING

In this section we discuss the TM training and explore different design choices for the head model: head architecture type and size, sequence scaling, model parameterization $Y$, head batch size, and time weighting. In each of the following experiments we start from a base model and ablate on the specific design choice. Our **base model** is: a medium size head, with sequence scaling of 1, difference paramterization $Y = X_1 - X_0$, head batch size of 4, uniform backbone and head time weighting, and sampled with $32 \times 32$ backbone-head steps; FM uses 256 steps.

**Head architectural type** One natural design choice is the architecture type of the head model $u_{s|t}$. We considered three options, see also fig. 2: (i) **MLP** - This is the most basic choice, used in (Shaul et al., 2025), where an MLP acting independently on each image token $y^i$ given the current step's latent $h_t$, and producing a prediction in token dimension $\mathbb{R}^d$, i.e., $u_{s|t}(y^i|h_t^i) \in \mathbb{R}^d$, $i = 1, \ldots, n$. (ii) **Convolution** - we use 2D convolution layers across the image tokens $y^i$ with kernels of size $3 \times 3$. (iii) **Transformer**- incorporating attention layers across the image tokens $y^i$. Both Convolution and Transformer head architectures take in the entire sequence of tokens $y \in \mathbb{R}^{n \times d}$ and produce a prediction in the same dimension, i.e., $u_{s|t}(y|h_t) \in \mathbb{R}^{n \times d}$. For the transformer head we further apply $16 \times 16 \times 16 \rightarrow 8 \times 8 \times 64$ reshape layer to allow for equivalently efficient head.

**Head model size** To check the influence of both head type and size, we have experimented with a variety of head model sizes for each type: *x-small*, *small*, *medium*, *large*, and *x-large*. In particular these correspond to head models with hidden dimensions of $d_h \in \{768, 1024, 1280, 1536, 2048\}$ (remember that our backbone hidden dim is $d_h = 2048$) and $\{6, 6, 8, 12, 16\}$ layers, respectively. Figure 3 shows, for each architecture type, the model rank as a function of the relative additional parameters to the backbone. For D-TM this corresponds to the relative size of the head, while for flow matching (FM) we also show what happens when we add the same number of parameters to the backbone as our *medium* size head. Dense shows the affect of using the backbone for both backbone and head similar to Zhang et al. (2025). Note that while having a head considerably improves the model's rank, the size of the head does not show strong correlation with performance, even in the limit case with a head almost the size of the backbone (>1 relative size). Figure 3 (b) and (c) show the effect of different heads on inference and training iteration time. As expected both inference time and training time increase with the head size and is particularly costly for dense inference. To summarize, smaller head sizes already provide good performance, improved inference time compared to FM and roughly equivalent training time.

**Sequence scaling**   We tested the affect of scaling the number of tokens inserted into the head. To that end we trained three learnable linear layers (see fig. 10 (a)): $L_{\text{in},y} : \mathbb{R}^d \to \mathbb{R}^{l \times d}$, $L_{\text{in},h} : \mathbb{R}^{d_b} \to \mathbb{R}^{l \times d_b}$, and $L_{\text{out},y} : \mathbb{R}^{l \times d} \to \mathbb{R}^d$, where $L_{\text{in},y}$ maps each token $y^i$ into $l$ tokens of the same dimensions; $L_{\text{in},h}$ maps backbone latents $h^i$ into $l$ tokens of the same dimensions; and $L_{\text{out},y}$ maps back $l$ output tokens into a single latent token $y^i$. We experimented with different scalings $l \in \left\{1, 2^2, 3^2, 4^2, 5^2, 6^2\right\}$ where $\sqrt{l}$ is applied to each dimension (of size $k$) of the latent image $y \in \mathbb{R}^{k^2 \times d}$ or $h_t \in \mathbb{R}^{k^2 \times d_b}$. The sequence scaling law is incorporated in the head as follows

$$L_{\text{out},y} u_{s|t}(L_{\text{in},y} y \mid L_{\text{in},h} h_t). \tag{11}$$

Figure 4 shows that for the transformer head, scaling the head sequence improves the model ranking significantly. In contrast, for the MLP head, scaling up the sequence does not impact performance consistently. One possible explanation is that MLP is applied on each scaled token independently, compared to the transformer that share information across all tokens via the attention layers. In fig. 4 (b) we report the inference speed as a function of sequence scaling, and in (c) we show the affect on training speed. Notably, while the affect of sequence scaling is limited in inference speed it is rather significant during training.

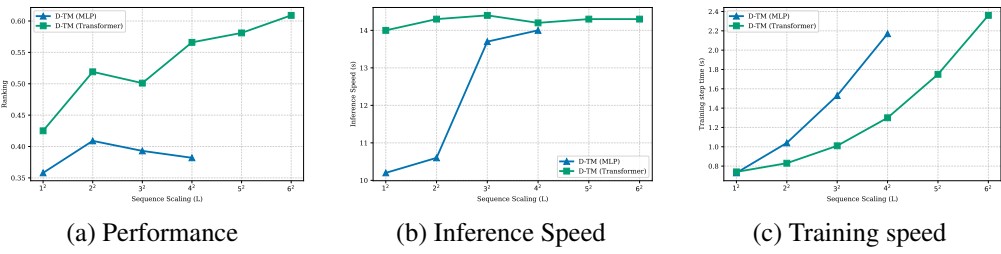

| (a) Performance | (b) Inference Speed | (c) Training speed |
| --- | --- | --- |

Figure 4: (a): Performance as a function of the sequence scaling factor. Scaling the input sequence to the head improves Transformer heads while not affecting MLP heads.

**Model parameterization $Y$**   Another design choice of the TM head is the choice of posterior $Y$ learned by the head and used to sample the next state $X_{t'}$. For the linear process (eq. (3)) formulating the two equations for $X_{t'}$ and $X_t$ results in three unknowns, $X_0, X_1, X_{t'}$, and a single known quantity, $X_t$. Therefore predicting one more quantity such as $X_0$ or $X_1$ or any independent relation of those two, would allow us to compute $X_{t'}$. Alternatively, we can also predict $X_{t'}$ directly but as mentioned above that would force us to introduce a second time parameter $t'$. Therefore, here we opt to experiment with the following options:

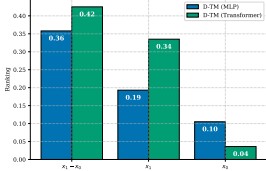

Figure 5: Performance of different $Y$ choices.

$$Y \in \{X_1 - X_0, X_1, X_0\} \qquad Y\text{-TM parameterizations} \tag{12}$$

Calculating $X_{t'}$ given $X_t$ and $Y$ gives an instantiation of the sampling relation in eq. (5) and given for completeness in appendix H. In fig. 12 we log the effect of different $Y$ parameterizations on the ranking of $Y$-TM models, where the difference parameterization $Y = X_1 - X_0$ is better than denoiser $Y = X_1$ which is much better than noise prediction $Y = X_0$. In appendix G we show the ablation of the flow matching target used in the FM loss, where the difference parameterization (used in eq. (9)) is also favorable.

**Head batch size**   Another simple and moderately effective scaling law can be achieved by increasing the batch size used by the head (see fig. 10 (b)). In practice for each time $t$ and state sample $X_t$, we consider head batch sizes $k_h \in \{1, 4, 16, 64\}$, see more details in appendix C. For the MLP architecture, since each token is processed independently (see fig. 2 second from left), we use different i.i.d. $s$

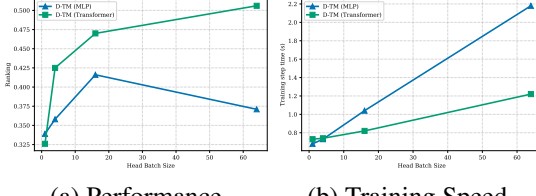

| (a) Performance | (b) Training Speed |
| --- | --- |

Table 1: (a) Performance as a function of the head batch size. (b) Training iteration time (seconds).

samples for each token; we name this time-per-token (TPT). In table 1 (a) we report experiments

comparing different head batch sizes for D-TM; we see that using a larger head batch size boosts performance, while reaching plateau after ~16 for the Transformer head. The impact of head batch size is smaller for the MLP head compared to Transformer, which can potentially be explained by the fact that MLP attends to each token as an independent sample, compared to a transformer where all patches of the same image are treated as a single sample. In table 1 (b) we report training speed, which shows that going beyond $k_h = 16$ might lead to significantly slower training times.

**Time weighting** A useful training design choice explored in (Esser et al., 2024) for flow matching is time weighting during training. In our case, instead of uniform time sampling $t, s \sim U(0, 1)$ in the TM loss (eq. (9)), we consider a non-uniform time sampling that will concentrate on certain times for the backbone ($t$) and head ($s$). Esser et al. (2024) noticed that for flow matching time $t$ it is beneficial to use a centered distribution favoring middle times in the interval $(0, 1)$. In particular, the log-normal distributions $\pi_{\ln}(\mu, \sigma)$ where $(\mu, \sigma) = (0, 1)$ was favorable. In our case, we tested time weighting for both the backbone $t$ and head $s$ time parameters for D-TM. We tested two $t$-weighting distributions similar to

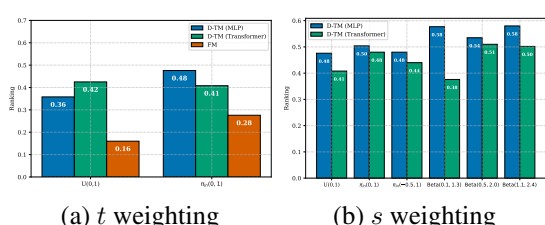

(a) $t$ weighting   (b) $s$ weighting

Table 2: Performance as a function of the backbone and head time weighting distribution. (a) Backbone time weighting ($t$), with Uniform time weighting on head. (b) Head time weighting ($s$), with lognormal time weighting on backbone.

flow matching, $U(0, 1)$ and $\pi_{\ln}(0, 1)$, while for $s$-weighting we tested $U(0, 1)$, $\pi_{\ln}(0, 1)$, $\pi_{\ln}(-0.5, 1)$ and $\text{Beta}(0.1, 1.3)$, $\text{Beta}(0.5, 2.0)$, and $\text{Beta}(1.1, 2.4)$, where Beta denotes the beta distributions. We chose the values to cover different $s$-time profiles, see fig. 11 for an illustration. We report results for the ablations of the backbone time $t$ weighting in table 2 (a) and for the head time $s$ (with $\pi_{\ln}(0, 1)$ for backbone) in (b). In general, backbone training enjoys standard log-normal time weighting $\pi_{\ln}(0, 1)$, although the transformer head is equally good with uniform time weight. For the head time weighting, both Beta and standard log-normal works well with the exception of $\text{Beta}(0.1, 1.3)$ for the Transformer head.

## 3.3 INFERENCE

We move to discuss design choices in sampling of the models at inference time, i.e., once training is done. We focus on efficiency-quality tradeoffs and investigate the affect of a novel stochastic sampling method that, under a particular design and hyperparameter regime, offers a significant quality boost at no extra sampling computational cost to D-TM.

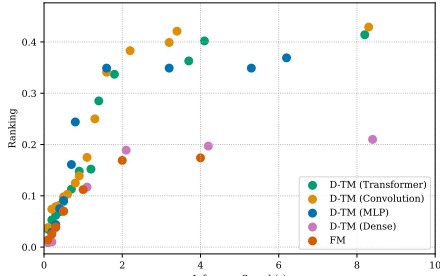

Figure 6: Performance as a function of Inference speed. Each color represents a different model, and each dot represents a different setup of transition steps and head NFE. We show Pareto optimal dots.

**Efficiency-quality tradeoff** When sampling a D-TM model we have the degree of freedom of choosing the $t$ and $s$ time discretizations both affecting the total (wall-clock) generation time. Throughout this experiment we keep $s$ and $t$ equidistant, i.e., $t = i/T$ and $s = j/S$ where $T, S$ are number of steps we ablate over. Figure 6 is a scatter plot showing inference wall-clock speed (in seconds) versus model rank for a collection of $T, S$, namely we consider $T \in \{1, 2, 4, 8, 16, 32, 64, 128, 256\}$ and $S \in \{1, 2, 4, 8, 16, 32\}$. We compare flow matching (FM), and D-TM with the three head architectures (MLP, Convolution, and Transformer), as well as the Dense architecture. As can be seen in the figure, D-TM (MLP, Transformer, and Convolution) sampling can be made both faster and of higher quality. For example, FM peak performance is with 32 midpoint samples (64 NFEs, corresponding to the 4sec red dot), where D-TM-MLP can achieve higher ranking with 0.8 second providing a $\sim 5\times$ wall-clock speedup.

**Stochastic samplers** In this last section we show that a certain family of stochastic samplers provides significant improvements to D-TM sample quality at no additional computational cost over standard TM sampling. The basic observation is that we can sample our trained model using any given supervision process $\tilde{q}$ as long as: (i) it has the same marginal distributions as the one we trained our model with $q$; and (ii) $\tilde{q}_{t'|t,Y}(\cdot|X_t, Y)$ is known and can be sampled efficiently. Inspired by (Xu et al., 2023b) we develop a family of stochastic samplers for D-TM for the case of Gaussian source noise, i.e., $p_0 = \mathcal{N}(0, I)$. For Gaussian noise the conditional probability path takes the

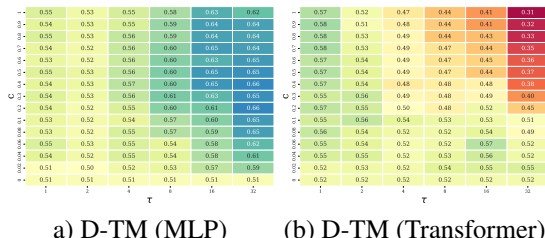

a) D-TM (MLP)  (b) D-TM (Transformer)

Table 3: Stochastic sampling performance (i.e., model rank) for various $c, \tau$ for the D-TM with MLP head in (a), D-TM with Transformer head in (b). Red colors indicate low ranking, while blue correspond to high ranking.

form $q_t(x|x_1) = \mathcal{N}(x|tx_1, (1-t)^2 I)$. Now, let $0 \le t < t' < t'' \le 0$ be three consecutive times, then given a sample $X_{t''} \sim q_{t''}(\cdot|X_1)$ we can use it to sample from the marginal $q_{t'}$ by (see appendix B for details)

$$X_{t'} = \frac{1}{t''}\left(t' X_{t''} + Z\right), \text{ with } Z \sim \mathcal{N}\left(0, (t'' - t')(t' + t'' - 2t't'')I\right). \tag{13}$$

Note that the joint probability of $(X_{t'}, X_{t''})$ is no longer the same as the supervision process $q$ but shares its marginals at times $t', t''$; where in the extreme case of $t'' = 1$ we get that $X_{t'} \perp X_{t''}$ (independent). This allows introducing more stochasticity into the D-TM sampling process where we explore two hyperparameters: (i) *scale* $c \in [0, 1]$ used to set $t'' = t' + c(1 - t')$; and (ii) *frequency* $\tau \in \{1, 2, \dots, T\}$ setting how often should we add a stochastic step; see algorithm 1 for the pseudocode of the stochastic D-TM sampling. Intuitively, the algorithm uses the D-TM prediction of $Y = X_1 - X_0$ to move to a future state $X_{t''}$ at time $t''$, and then adds independent noise at the right amount so to achieve a sample $X_{t'}$ at the earlier time $t'$. In this experiment we tested $32 \times 32$ backbone-head sampling, which gave near optimal performance in table 3, and the following hyper-parameters

---

**Algorithm 1** Stochastic D-TM sampling.

**Require:** $p_{Y|t}^\theta$         ▷ Trained model
**Require:** $T$         ▷ Backbone steps
**Require:** $c, \tau$         ▷ Scale and frequency
1: Sample $X_0 \sim \mathcal{N}(0, I)$
2: **for** $t = 0, \frac{1}{T}, \frac{2}{T}, \dots, 1 - \frac{1}{T}$ **do**
3:      Sample $Y \sim p_{Y|t}^\theta(\cdot|X_t)$
4:      $t' \leftarrow t + \frac{1}{T}$
5:      **if** $t \pmod{\lceil T/\tau \rceil} = 0$ **then**
6:         $t'' \leftarrow t' + c(1 - t')$
7:         Compute $X_{t''}$     ▷ eq. (7)
8:         Compute $X_{t'}$     ▷ eq. (13)
9:      **else**
10:        Compute $X_{t'}$     ▷ eq. (7)
11:      **end if**
12: **end for**
13: **return** $X_T$

---

$(c, \tau) \in \{0, 0.02, 0.04, \dots, 0.1, 0.2, \dots, 1\} \times \{1, 2, 4, 8, 16, 32\}$. Table 3 (a) and (b) show the affect of stochastic sampling on D-TM MLP and Transformer head (resp.). Interestingly, MLP head enjoys the stochastic sampling more that Transformer head leading to a 0.66 rank (+0.15 from standard sampling), which is the highest in our experiments, and is consistently achieved across high frequency sampling. Transformer head reaches its peak performance of 0.58 rank (+0.06) with low frequency sampling. In appendix I we discuss applying algorithm 1 to flow matching.

## 3.4 SUMMARY RESULTS FOR SELECTED MODELS

Each of the design space exploration above tested ablations from a base model (detailed in section 3.2). Among all these ablations we picked two design choices balancing performance and train/inference speed: (i) D-TM with MLP head, head batch size $k_h = 16$, lognormal $s$ and $t$ time weights, with and without high frequency stochastic sampling $\tau = 32$, $c = 0.2$; and (ii) D-TM with Transformer head, sequence scale $l = 4$, head batch size $k_h = 16$, lognormal $s$ and $t$ time weights, with and without low frequency stochastic sampling $\tau = 1$, $c = 0.8$. In table 4 and figs. 1 and 7 to 9 we present our most performing D-TM variants discovered via the previous experiments compared with relevant baselines implemented, trained and evaluated under the exact same setting. The baselines include: the D-TM variant in (Shaul et al., 2025) and the Dense version in (Zhang et al., 2025); Flow

Matching (FM) with its optimal time-weighting variant (Esser et al., 2024); Autoregressive image generation with a flow matching head generating continuous tokens (AR/MAR) Team et al. (2025); Li et al. (2024); Fan et al.; discrete tokens autoregressive (AR) (Yu et al., 2022); and discrete tokens masked autoregressive (MAR) (Chang et al., 2022). Due to lack of space we only present subset of the metrics where all the data, including all evaluations are presented in appendix L. As can be seen in the table, design choice (i) with MLP head and high frequency sampling (denoted DTM++) reaches the top performance with 0.66 rank score; and design choice (ii) includes the Transformer head and excels at image quality (see e.g., Aesthetic and PickScore) but do not benefit as much from the stochastic sampling and ends up being second to the MLP head (denoted DTM+). As sequence scaling is training-costly we limited it to $l = 4$, note that for $l = 36$ (see Sequence scaling) the Transformer head model becomes competitive with our best model.

Table 4: Main Results comparing most performant D-TM variants and relevant baselines.

| Model | Head | | | | | Sampling | MS-COCO | | | | PartiPrompts | | | | GenEval | T2ICompBench | |
| | Type | Size | Seq scale | batch size | Weighting | | CLIPScore ↑ | PickScore ↑ | Aesthetic ↑ | ImageReward ↑ | CLIPScore ↑ | PickScore ↑ | Aesthetic ↑ | ImageReward ↑ | Overall ↑ | Overall ↑ | Rank ↑ |
|---|---|---|---|---|---|---|---|---|---|---|---|---|---|---|---|---|---|
| DTM | MLP | mid | 1 | 4 | $U(0,1) \times U(0,1)$ | linear | 26.2 | 21.3 | 5.64 | 0.28 | 26.6 | 21.2 | 5.46 | 0.51 | 0.55 | 0.4422 | 0.36 |
| DTM | MLP | mid | 1 | 16 | $\pi_{ln}(0,1) \times \pi_{ln}(0,1)$ | linear | 26.4 | 21.4 | 5.69 | 0.4 | 27.0 | 21.3 | 5.47 | 0.63 | 0.55 | 0.4549 | 0.51 |
| DTM++ | MLP | mid | 1 | 16 | $\pi_{ln}(0,1) \times \pi_{ln}(0,1)$ | $c = 0.2, \tau = 1$ | 26.3 | 21.5 | 5.78 | 0.47 | 27.0 | 21.4 | 5.57 | 0.70 | 0.58 | 0.4625 | 0.66 |
| DTM | Convolution | mid | 1 | 4 | $U(0,1) \times U(0,1)$ | linear | 26.1 | 21.4 | 5.76 | 0.32 | 26.4 | 21.3 | 5.52 | 0.51 | 0.54 | 0.4294 | 0.4 |
| DTM | Transformer | mid | 1 | 4 | $U(0,1) \times U(0,1)$ | linear | 26.1 | 21.4 | 5.76 | 0.31 | 26.5 | 21.3 | 5.54 | 0.51 | 0.54 | 0.434 | 0.43 |
| DTM | Transformer | mid | 4 | 16 | $\pi_{ln}(0,1) \times \pi_{ln}(0,1)$ | linear | 26.2 | 21.6 | 5.87 | 0.44 | 26.6 | 21.4 | 5.59 | 0.62 | 0.50 | 0.4461 | 0.52 |
| DTM+ | Transformer | mid | 4 | 16 | $\pi_{ln}(0,1) \times \pi_{ln}(0,1)$ | $c = 0.8, \tau = 32$ | 26.2 | 21.6 | 5.88 | 0.44 | 26.6 | 21.4 | 5.58 | 0.63 | 0.58 | 0.4487 | 0.58 |
| DTM | Dense | | | | $U(0,1) \times U(0,1)$ | linear | 25.9 | 21.3 | 5.69 | 0.18 | 26.1 | 21.2 | 5.44 | 0.36 | 0.52 | 0.4185 | 0.25 |
| FM | | | | | $U(0,1)$ | linear | 25.9 | 21.2 | 5.55 | 0.14 | 26.1 | 21.1 | 5.33 | 0.34 | 0.50 | 0.4252 | 0.16 |
| FM | | | | | $\pi_{ln}(0,1)$ | linear | 26.2 | 21.3 | 5.67 | 0.3 | 26.6 | 21.2 | 5.44 | 0.48 | 0.52 | 0.4332 | 0.28 |
| AR | | | | | | argmax | 26.7 | 20.3 | 4.93 | -0.06 | 26.7 | 20.4 | 4.81 | -0.01 | 0.41 | 0.3879 | 0.17 |
| AR | MLP | mid | 1 | 4 | $U(0,1)$ | linear | 24.8 | 20.1 | 4.76 | -0.48 | 24.9 | 20.1 | 4.5 | -0.43 | 0.34 | 0.3429 | 0.08 |
| MAR | | | | | | argmax | 26.6 | 20.6 | 5.27 | 0.01 | 26.8 | 20.7 | 5.15 | 0.14 | 0.44 | 0.3944 | 0.19 |
| MAR | MLP | mid | 1 | 4 | $U(0,1)$ | linear | 26.1 | 20.7 | 5.06 | 0.17 | 27.0 | 20.7 | 4.95 | 0.33 | 0.52 | 0.4393 | 0.19 |

## 4 RELATED WORK

Iterative generative models, which gradually transform noise to data, were pioneered with diffusion models (Sohl-Dickstein et al., 2015; Ho et al., 2020) and further generalized and improved with flow matching (Lipman et al., 2022; Liu et al., 2022; Albergo & Vanden-Eijnden, 2022). Transition Matching (TM) (Shaul et al., 2025) is a further recent generalization that replaces the simplified transition kernel in diffusion and flow models with a more expressive internal generative model. A related method that used the same difference modeling but with a single backbone architecture (called Dense in this paper) was developed in Zhang et al. (2025). TM uses head-backbone construction to inject relevant inductive bias and maintain efficiency; similar head MLP constructions was introduced and incorporated in continuous autoregressive image generation methods (Li et al., 2024; Fan et al.; Team et al., 2025), however different head models and architectures were not systematically explored previously. Similar to our work, systematic design space exploration was done for diffusion models in Karras et al. (2022). Stochastic sampling has shown some benefit in diffusion (Song et al., 2020) and flow (Ma et al., 2024) models while usually based on adding Langevin dynamics to existing SDE/ODE formulation. Our TM stochastic sampler is inspired by Xu et al. (2023b) that re-noises the variance exploding denoiser in Karras et al. (2022), see more details in the Stochastic samplers section.

## 5 CONCLUSIONS

We conducted a large-scale ablation of design choices in continuous-time Transition Matching, focusing on the D-TM variant. Our experiments highlight that head design choices—architecture, size, sequence scaling, head batch size, and time scheduling — can significantly improve generative quality and efficiency, even with a fixed backbone and dataset. Some design choices, such as large sequence scaling and head batch size improve performance but at the cost of increase training times and/or memory footprint. Stochastic TM sampling offers a further significant improvement in generation quality at not extra computational cost. It is not clear to the authors why token-wise head (i.e., MLP) leads to improved text-adherence scores compared to the more expressive Transformer head, which opens an interesting future research question. Furthermore, we emphasize that the conclusions of this paper are based on a focused study of image generation at 256 resolution, where higher resolution images and/or other media modalities such as video and audio could potentially lead to different conclusions and provide valuable future work direction. LLMs were used in this paper to aid/polish writing in introduction and abstract sections.

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

# A  QUALITATIVE RESULTS

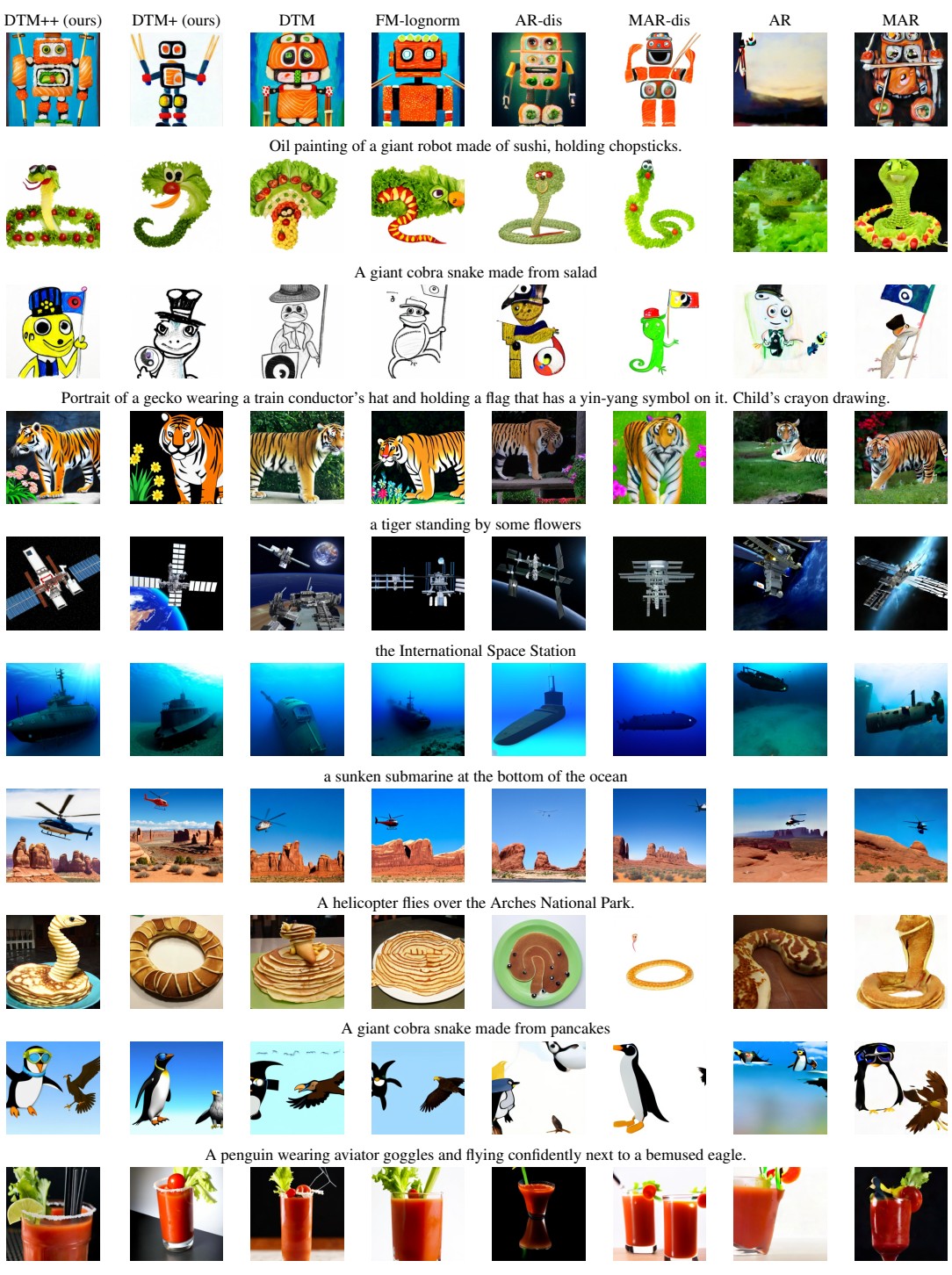

Figure 7: (Cont.) Samples comparing of our Best D-TM MLP (DTM++) and Transformer (DTM+) Models with DTM baseline, FM-lognormal, AR, MAR, AR-discrete, MAR-discrete as baselines. All models share similar architecture and training recipe.

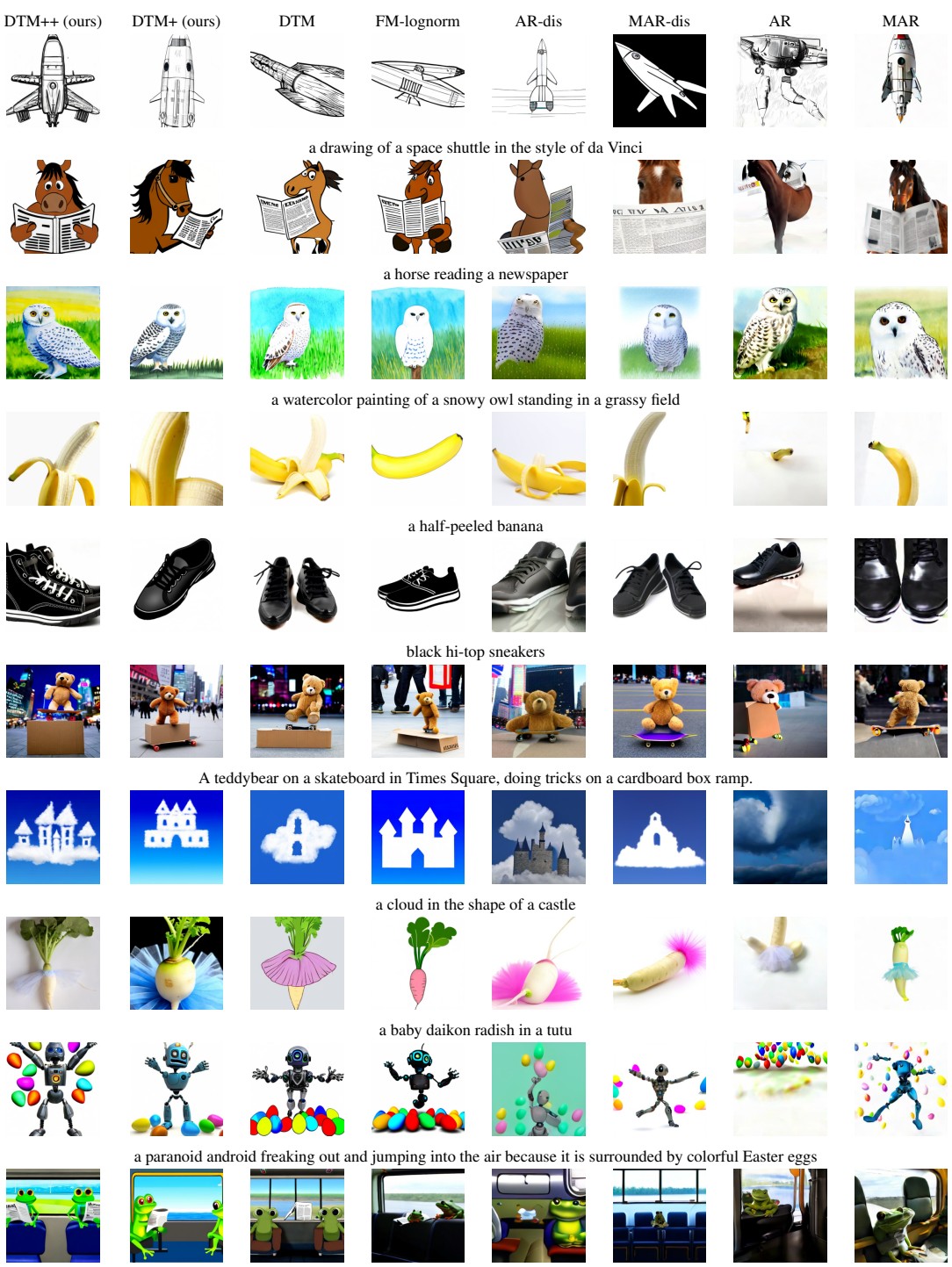

Figure 8: (Cont.) Samples comparing of our Best D-TM MLP (DTM++) and Transformer (DTM+) Models with DTM baseline, FM-lognormal, AR, MAR, AR-discrete, MAR-discrete as baselines. All models share similar architecture and training recipe.

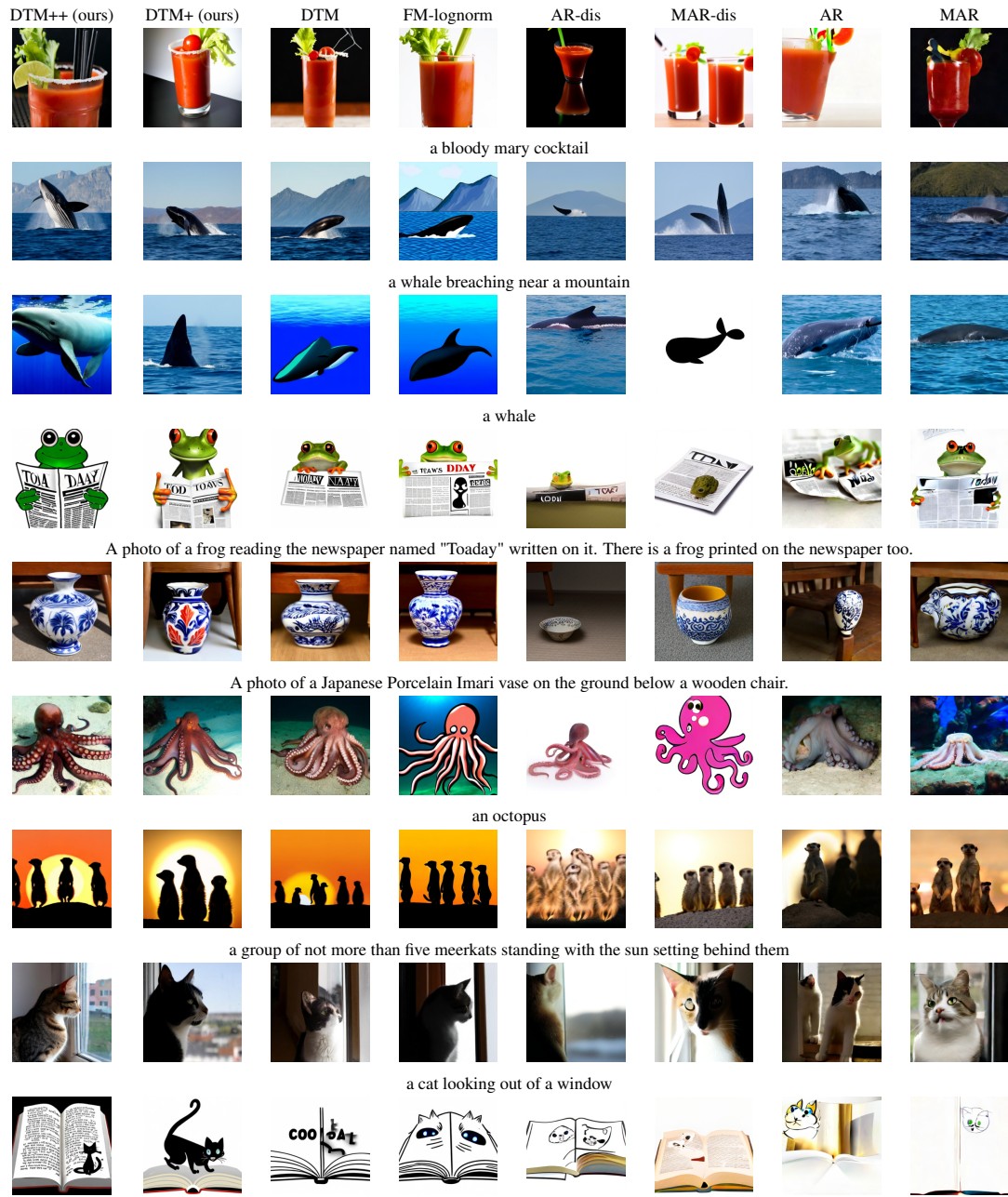

Figure 9: (Cont.) Samples comparing of our Best D-TM MLP (DTM++) and Transformer (DTM+) Models with DTM baseline, FM-lognormal, AR, MAR, AR-discrete, MAR-discrete as baselines. All models share similar architecture and training recipe.

## B  STOCHASTIC SAMPLING

We derive eq. (13) used in the D-TM stochastic sampling in algorithm 1. consider two arbitrary times $0 \leq a < b \leq 1$ in the $[0, 1]$ time interval, and $x_1$ a constant data sample. Let $X_b$ be a sample from the conditional probability path $q_b$ at time $b$, i.e.,

$$X_b \sim \mathcal{N}(\cdot \,|\, bx_1, (1-b)^2 I). \tag{14}$$

Now, we want to transform $X_b$ to a sample $X_a$ from the conditional probability path at time $a$, i.e.,

$$X_a \sim \mathcal{N}(\cdot \,|\, ax_1, (1-a)^2 I). \tag{15}$$

To that end note that

$$\frac{a}{b} X_b \sim \mathcal{N}\left(\cdot \,\bigg|\, ax_1, \left(\frac{a(1-b)}{b}\right)^2 I\right). \tag{16}$$

Now let $Z \sim \mathcal{N}(\cdot \,|\, 0, \sigma^2 I)$ be an independent Gaussian sample with $\sigma > 0$ as a degree of freedom. Therefore

$$\frac{a}{b} X_b + Z \sim gN\left(\cdot \,\bigg|\, ax_1, \left[\left(\frac{a(1-b)}{b}\right)^2 + \sigma^2\right] I\right). \tag{17}$$

Lastly, we solve for $\sigma$ such that

$$\left(\frac{a(1-b)}{b}\right)^2 + \sigma^2 = (1-a)^2 \tag{18}$$

leading to

$$\sigma^2 = \frac{(b-a)(a+b-2ab)}{b^2}. \tag{19}$$

This coincides with eq. (13) if we set $b = t''$ and $a = t'$.

## C  HEAD BATCH SIZE

The loss corresponding for head batch size $k_{\mathrm{h}} \in \{1, 4, 16, 64\}$ takes the form

$$\mathcal{L}(\theta) = \mathbb{E}_{t, X_t, s_i, Y_{0,i}} \frac{1}{k_{\mathrm{h}}} \sum_{i=1}^{k_{\mathrm{h}}} \left\| u_{s_i|t}^{\phi}(Y_{s_i} | f_t^{\varphi}(X_t)) - (Y_1 - Y_{0,i}) \right\|^2, \tag{20}$$

where $s_i \sim U(0, 1)$ are random i.i.d., $Y_{0,i} \sim \mathcal{N}(0, I)$, and $Y_{s_i} = (1 - s_i) Y_{0,i} + s_i Y_1$ for $i = 1, \ldots, k_{\mathrm{h}}$.

# D ILLUSTRATIONS

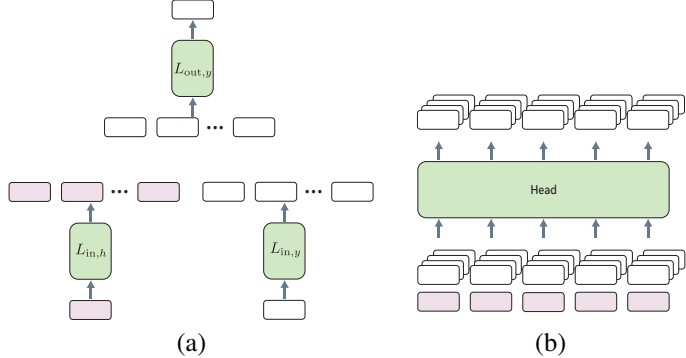

Figure 10: (a) Scaling the sequence length entering the head. (b) Scaling the head batch size.

## E    TIME WEIGHTING DISTRIBUTIONS

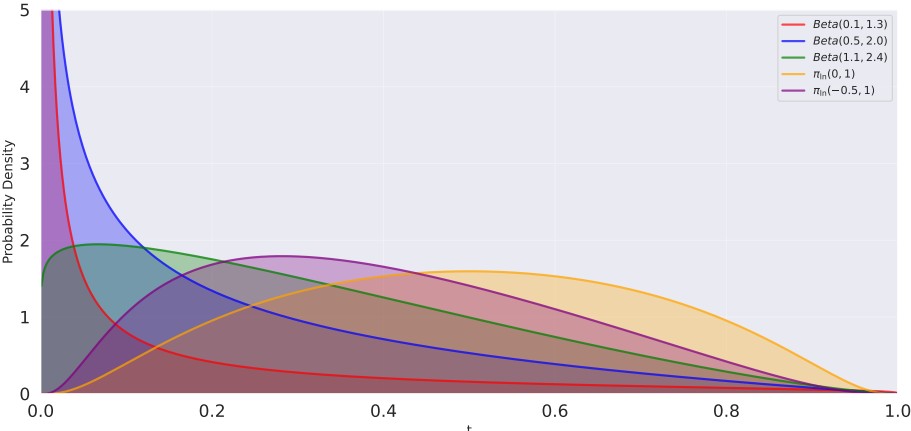

Figure 11: Visualization of the Beta and lognormal distributions used for time weighting.

## F    CLASSIFIER FREE GUIDANCE

We use Classifier Free Guidance (CFG) (Ho & Salimans, 2022), defined by a weight $\omega > 0$ and the corresponding transformation to the velocity (Zheng et al., 2023),

$$\tilde{u}_{s|t}(\cdot|h_{t,c}) = (1 - \omega)u_{s|t}(\cdot|h_{t,c}) - \omega u_{s|t}(\cdot|h_{t,\emptyset}), \tag{21}$$

where we use the standard choice of $w = 6.5$ and our latent representation is learned with a condition $C = c$, e.g., text prompts, that can also be empty, denoted by $C = \emptyset$,

$$h_{t,C} = f_t^{\varphi}(X_t, C). \tag{22}$$

In the training loss (eq. (9)) we random a prompt-image pair from the dataset, $(C, X_1)$, where with probability 0.15 we set $C = \emptyset$.

## G    FLOW MATCHING HEAD PARAMETERIZATION

Another design choice for the TM head is the target predicted by the head model used to sample $Y$ (i.e., the target in the FM loss eq. (9)). As known in flow matching and diffusion literature (Lipman et al., 2024) one can choose different targets to sample $Y_1 = Y \sim p_{Y|t}(\cdot|X_t)$ (and change the sampling in eq. (10) accordingly). Flow matching learns the difference $Y_1 - Y_0$; the denoiser prediction learns $Y_1$ and noise prediction $Y_0$. Figure 12 (b) shows comparison of these parameterizations for $Y = X_1 - X_0$ where in essence: the difference target $Y_1 - Y_0$ is slightly better than denoiser $Y_1$ which is considerably better than noise prediction $Y_0$. We set the difference $Y_1 - Y_0$ as in eq. (9) as our default choice.

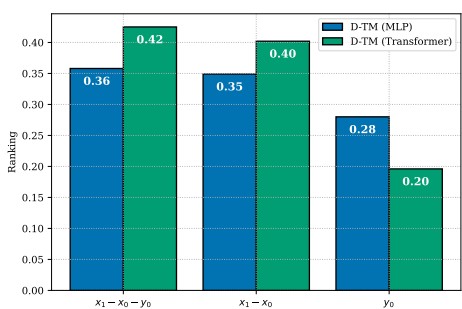

Figure 12: Performance of different flow matching targets.

## H $Y$ SAMPLING RELATIONS

Calculating $X_{t'}$ given $X_t$ and $Y$ can be done by writing the equations of the linear process (in eq. (3)) for $t$ and $t'$ and solving for $X_{t'}$ as functions of $Y$ and $X_t$ leading to an instantiation of the sampling relation in eq. (5) for the $Y$ choices in eq. (12),

$$X_{t'} = X_t + (t' - t)Y \qquad \text{with } Y = X_1 - X_0 \quad \textit{(Difference)} \qquad (23)$$

$$X_{t'} = \frac{(1 - t')X_t + (t' - t)Y}{1 - t} \qquad \text{with } Y = X_1 \qquad \textit{(Denoiser)} \qquad (24)$$

$$X_{t'} = \frac{t'X_t + (t - t')Y}{t} \qquad \text{with } Y = X_0 \qquad \textit{(Noise)} \qquad (25)$$

## I APPLYING TM STOCHASTIC SAMPLING TO FM

As a side contribution, we found that applying algorithm 1 where sampling with eq. (7) is replaced with standard ODE solve in FM also leads to considerable gains in FM sampling, specifically for medium frequency $8 - 16$ (for $T = 32$ NFEs) and scale $c > 0.1$, see fig. 13. Note that this algorithm is similar to the restart algorithm in Xu et al. (2023b) with several key differences: we use if for flow matching rather than variance exploding denoiser EDM, we parameterize it with two simple parameters, and we avoid the EDM discretization scheme. Lastly, note that for FM sampling this algorithm increases computational cost as it requires ODE solving during the sampling algorithm, this is in contrast to using it in TM which does not increase computational cost.

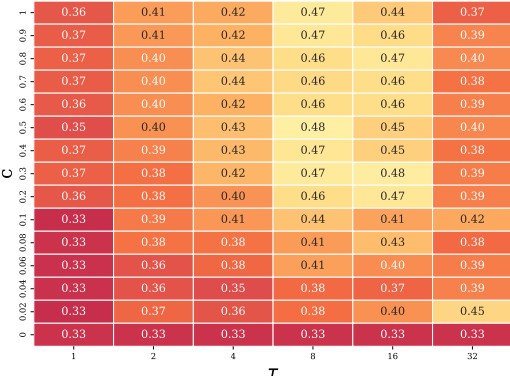

Figure 13: Applying a version of the TM stochastic sampling algorithm (algorithm 1) to flow matching, where eq. (7) is replaced with ODE sampling (and consequently requires more NFE), seems to also improve FM generation ranking. Red colors indicate low ranking, while blue colors correspond to high ranking; colors are on consistent scale with table 3.

## J  RELATION OF $Y$ PARAMETERIZATION TO DIFFUSION AND FLOW MATCHING

Predicting $Y = X_1 - X_0$ can be seen as the TM version of flow matching (Shaul et al., 2025), which instead predicts the *deterministic function* $\mathbb{E}[X_1 - X_0 | X_t = x_t]$; $Y = X_1$ is the TM versions of a denoiser (Salimans & Ho, 2022) which predicts the function $\mathbb{E}[X_1 | X_t = x_t]$, while $Y = X_0$ is the TM version of noise-prediction Ho et al. (2020) which predicts the function $\mathbb{E}[X_0 | X_t = x_t]$. In contrast to flow matching or diffusion, TM learns to sample from the posterior $X_1 - X_0$, $X_1$ or $X_0$ directly rather than estimating their mean as done in flow matching and diffusion models. Note that similarly to the situation with denoiser and noise-prediction (see e.g., (Lipman et al., 2024)), also for $Y$-TM, the former has a singularity near $t = 1$ and the latter near $t = 0$. The singularity near $t = 0$ becomes an issue in sampling as numerical instability is introduced at the beginning of the sampling rather at the end, providing a potential explanation to the lower performance of the $Y = X_0$ parameterization.

## K    D-TM SAMPLING PSEUDOCODE

For completeness we provide the standard D-TM sampling pseudocode for continuous time in algorithm 2.

---

**Algorithm 2** DTM Sampling adapted from Shaul et al. (2025) for continuous time.

---

**Require:** Trained model $(u^\phi, f^\varphi)$
**Require:** Time grids $0 = t_0 < t_1 < \cdots < t_T = 1$ and $0 = s_0 < s_1 < \cdots < s_S = 1$.
 1: Sample $X_0 \sim \mathcal{N}(0, I_d)$
 2: **for** $i = 0$ **to** $T - 1$ **do**
 3:     $h_i \leftarrow f_{t_i}^\varphi(X_i)$
 4:     Sample $Y_0 \sim N(0, I)$
 5:     $Y \leftarrow \texttt{ode\_solve}(Y_0, u_{\cdot|t_i}^\phi(\cdot|h_i), \{s_0, \ldots, s_S\})$
 6:     $X_{i+1} \leftarrow X_i + \frac{1}{T}Y$
 7: **end for**
 8: **return** $X_T$

---

# L  FULL RESULTS

Table 5: All ablations and evaluated models used in the paper.

Table 6: (Cont.) All ablations and evaluated models used in the paper.

