# OpenReview forum: "Exploring the Design Space of Transition Matching"
_ICLR.cc/2026/Conference — ICLR 2026 Poster_

### Official Review · Reviewer_yyoT · 2025-10-31

**Soundness:** 4
**Presentation:** 4
**Contribution:** 4
**Rating:** 6
**Confidence:** 4

**Summary:**

This paper presents a large-scale and systematic investigation into the design, training, and sampling of the head module in the continuous-time bidirectional variant of Transition Matching (TM) for generative modeling, with a focus on text-to-image generation. The work addresses a notable gap in existing literature— the lack of in-depth exploration of the head module’s architecture and hyperparameters— despite its crucial role in translating backbone latent representations into concrete transition outputs. By training 56 different 1.7B text-to-image models and conducting 549 unique evaluations, the authors provide comprehensive insights into how head design choices impact generation quality, training efficiency, and inference efficiency. The paper’s contributions, including the novel stochastic sampling algorithm and actionable guidelines for TM-based models, are valuable for advancing generative modeling research.

**Strengths:**

Rigorous Experimental Design
The authors ensure fair comparison across models and baselines by keeping the backbone architecture (a 1.7B parameter DiT transformer), training dataset (350M text-image pairs), and most training hyperparameters fixed. Evaluations are conducted on four datasets (MS-COCO, PartiPrompts, GenEval, T2ICompBench) using 25 metrics, which are aggregated into a single rank score— this comprehensive setup enhances the reliability and generalizability of the results.
 Comprehensive Exploration of Head Design Space
The paper systematically explores key design choices for the head module, including architecture type (MLP, Convolution, Transformer), size (x-small to x-large), sequence scaling, batch size, time weighting, and model parameterization (Y). This exploration reveals non-trivial findings, such as the observation that smaller head sizes can already deliver good performance without excessive computational cost, and that sequence scaling benefits Transformer heads (via attention-driven information sharing) but not MLP heads (which process tokens independently).

**Weaknesses:**

Lack of Analysis on Why MLP Heads Outperform Transformer Heads in Text Adherence
The paper notes that MLP heads (token-wise processing) achieve better text-adherence scores than more expressive Transformer heads, but does not provide a detailed analysis of the underlying reasons. Further investigation— such as analyzing token-level alignment between text prompts and generated images, or exploring how attention in Transformer heads might distract from text-image alignment— would strengthen the paper’s insights and open avenues for future research.

**Questions:**

Since the work focuses exclusively on text-to-image generation, how do you think your findings might generalize to other modalities like text-to-video or audio generation? Could you discuss potential adaptations for other modalities or highlight limitations specific to text-to-image?

---

> ### Author Response · Authors · 2025-11-30
>
> **Q:** Lack of Analysis on Why MLP Heads Outperform Transformer Heads in Text Adherence.
>
> **A:** Thank you for raising the point regarding the lack of analysis on why MLP heads outperform Transformer heads in text adherence. These are indeed good questions, as also finding theoretical grounding to other phenomena observed in the paper such as the effect of sequence scaling, model parameterization choice, time weighting and positive effect of stochastic sampling. While we agree on the importance of theoretical grounding, we want to highlight that the current theoretical understanding of **generalization in flow and diffusion models** is extremely limited and largely relies on empirical evidence (e.g., [1]). By choice, we decided to focus in this paper on providing the community with a comprehensive set of empirical data for the Transition Matching framework and draw some empirical observations out of these experiments. We strongly believe that such data-driven studies are necessary to establish new paradigms (if not theoretical understanding) and advance the field.
>
> *[1] On the Closed-Form of Flow Matching: Generalization Does Not Arise from Target Stochasticity, Bertrand et al. 2025.*\
> \
> \
> **Q:** How do you think your findings might generalize to other modalities like text-to-video or audio generation?
>
> **A:** Thank you for this question, in this paper we worked specifically with image modalities. Following the reviewers’ comments we added a sentence in the conclusion to scope our empirical findings to the image modality only. However, as a side note, although not included in the paper, note that we have since conducted preliminary experiments with larger models (up to 25B parameters), higher-resolution images (up to 1k), and other modalities such as video and audio. In all cases, our proposed methods rather consistently outperformed the baseline FM following this paper’s observations.
> However, to correctly encapsulate this paper’s empirical evidence we added the following sentence to the conclusions of our work:
> Furthermore, we emphasize that the conclusions of this paper are based on a focused study of image generation at 256 resolution, where higher resolution images and/or other media modalities such as video and audio could potentially lead to different conclusions and provide valuable future work direction.

---

### Official Review · Reviewer_fSgD · 2025-10-31

**Soundness:** 2
**Presentation:** 3
**Contribution:** 1
**Rating:** 4
**Confidence:** 5

**Summary:**

This paper presents a comprehensive study on design and training strategies for text-to-image generation models, focusing on how different head architectures, parameterizations, and scaling choices affect performance. The authors evaluate over 500 models across multiple datasets and 25 metrics, aggregating results into a unified ranking system. They find that simpler head architectures, such as MLP-based ones, can surprisingly outperform more complex transformer heads in text adherence, raising questions for future research. The study provides extensive ablation results and benchmarking insights relevant to improving model quality and efficiency in large-scale generative modeling.

**Strengths:**

- The primary contribution of this work is a large-scale, systematic exploration of the head module's design space within the Transition Matching (TM) framework. To my knowledge, this is the first paper to conduct such a comprehensive ablation study, systematically investigating the impact of head architecture (e.g., MLP, Transformer) , model size , sequence scaling , and batch size.

- While the paper's strength is its comprehensive empirical analysis (training 56 unique 1.7B models) , it also introduces a novel stochastic sampling algorithm for TM, which is shown to significantly improve generative quality at no extra computational cost.

- A key novel insight is the paper's clear and distinct analysis of the dual time-sampling parameters. It is the first to systematically decouple and study the effect of time weighting for both the backbone's transition time $t$ and the head's internal generative time $s$ during training. This is further complemented by an analysis of the corresponding $T$ and $S$ discretization steps at inference time.

- The extensive experiments provide actionable guidelines and valuable insights for the community regarding which design choices are most likely to yield improvements in quality and efficiency for this promising class of models

**Weaknesses:**

The paper's main contribution is a large-scale, systematic empirical study of the TM head's design space. While this comprehensive ablation is valuable and provides actionable insights, the work is light on fundamental algorithmic innovation. The paper does not introduce a new generative paradigm but rather exhaustively explores the design choices of an existing one. Although a novel stochastic sampler is presented, the paper's core feels more like an extensive experimental report than a proposal of a new, innovative method.

**Questions:**

A significant drawback of the Transition Matching paradigm, as implemented and studied here, is the extraordinarily high inference cost. The paper's architecture requires the large 1.7B-parameter Backbone network to be called $T$ times (e.g., 32 times), and crucially, the Head network to be called $T \times S$ times (e.g., $32 \times 32 = 1024$ times) per generation. This places TM models at a major computational disadvantage compared to conventional diffusion or flow models, which only require $T$ evaluations.

The inference speed reported in the paper (e.g., **21.3 seconds per sample** for Model, DTM MLP) is excessively slow for practical application, making the framework currently intractable for real-time or high-throughput generation. While the paper provides an exhaustive empirical analysis of the design space, it is regrettable that the authors did not dedicate effort to mitigating this core issue. The lack of proposed architectural or algorithmic solutions (e.g., knowledge distillation, progressive sampling optimization, or alternative ODE solvers) to tackle this major computational bottleneck is a key limitation of this work. We believe addressing the high latency of the $T \times S$ Head evaluations is the most pressing challenge for TM models.

---

> ### Author Response · Authors · 2025-11-30
>
> **Q:** A significant drawback of the Transition Matching paradigm, as implemented and studied here, is the extraordinarily high inference cost.
>
> **A:** This is in fact incorrect. We would like to draw your attention to Figure 6, which demonstrates that DTM not only surpasses FM in performance but also achieves approximately **five times faster inference speed**, without compromising results. This improvement is primarily due to two factors:
> 1. Each DTM step is more effective, allowing us to use significantly fewer backbone evaluations compared to FM to reach comparable performance.
> 2. The head network is lightweight, comprising only about 2% of the backbone’s parameters (and for the MLP head, even without attention mechanisms). Consequently, sampling from the head is substantially faster than from the backbone.

---

### Official Review · Reviewer_gvxM · 2025-10-31

**Soundness:** 2
**Presentation:** 2
**Contribution:** 2
**Rating:** 4
**Confidence:** 3

**Summary:**

Transition Matching (TM) is an emerging paradigm for generative modeling that generalizes diffusion, flow-matching, and continuous-state autoregressive models. TM gradually transforms noise into data samples, but it utilizes a second "internal" generative model to execute the transition steps, making these steps more expressive than in diffusion or flow models. The method employs a large backbone network and a smaller "head" module for efficiency. This paper presents a large-scale, systematic exploration of the TM design space, investigating critical components like network architecture, training objectives, and parameter configurations. The goal is to provide the community with in-depth insights and empirical guidance on how to efficiently design and deploy TM models.

**Strengths:**

1. Extensive and Systematic Experimental Exploration: The authors conducted a large-scale, systematic study, thoroughly investigating numerous design choices (architecture, loss functions, hyperparameters) within the Transition Matching framework, offering valuable empirical knowledge for this new domain.

2. Excellent Clarity and Writing: The paper is very well-written and clearly structured, articulating technical concepts and experimental findings effectively, which makes the core components and design trade-offs of this complex model paradigm easily digestible.

**Weaknesses:**

1. Limited Technical Novelty: The contribution leans more toward an empirical study that explores and summarizes existing design choices rather than introducing fundamental technical or algorithmic innovations.

2. Restricted Evaluation Scope: The training and testing are exclusively conducted on low-resolution images (256x256), which significantly undermines the reliability and generalizability of the findings for real-world applications where high-fidelity, high-resolution image generation is currently a major focus.

**Questions:**

None

---

> ### Author Response · Authors · 2025-11-30
>
> **Q:** Limited technical novelty: The contribution leans more toward an empirical study that explores and summarizes existing design choices rather than introducing fundamental technical or algorithmic innovations.
>
> **A:** While we appreciate your perspective on the technical novelty of our work, we feel that introducing and exploring new effective architectures and design choices, as well as sampling algorithms do qualify as technical novelty, and the progress in generative models very much relies on such advancements (see e.g., [2]).
>
> *[2] Scalable Diffusion Models with Transformers, Peebles and Xie, 2023.*\
> \
> \
> **Q:** Restricted Evaluation Scope: The training and testing are exclusively conducted on low-resolution images (256x256)...
>
> **A:** We acknowledge that our primary experiments were conducted on lower-resolution images, which may limit the generalizability of our findings. However, we view this focused scope as a strength, enabling many controlled comparisons across a wide range of design choices. As a side comment, we have conducted additional experiments of the methods proposed in this paper with larger models (up to 25B parameters), higher-resolution images (up to 1k), and other modalities such as video and audio. In all cases, our proposed methods consistently outperformed the baseline FM.
>
> To address the reviewer concern we added the following sentence to the conclusions of our work:
> Furthermore, we emphasize that the conclusions of this paper are based on a focused study of image generation at 256 resolution, where higher resolution images and/or other media modalities such as video and audio could potentially lead to different conclusions and provide valuable future work direction.

---

### Official Review · Reviewer_fGYX · 2025-10-31

**Soundness:** 4
**Presentation:** 4
**Contribution:** 4
**Rating:** 8
**Confidence:** 3

**Summary:**

The paper considers the question of optimal design of a Transition Matching (TM) model. In particular, the paper focuses on the design of the head module in TM. The authors perform a thorough exploration of the various choices pertaining to the head such as the architecture, model size, sequence scaling etc., The paper also presents "ideal" choices for the hyperparameters/choices considered wrt to the head module. In addition, the authors show, through some theoretical and experimental justification, a family of stochastic samplers that provide improved performance to D-TM models.

**Strengths:**

- The degree of thoroughness with which the authors designed the experiments and studied the various aspects of the head module is impressive and is presented clearly.
- Since TM is a relatively new and under-explored direction, such a study shedding some light on designing performant TM models is significant.

**Weaknesses:**

- There is not much discussion on the reasons for the behaviors observed in the experiments. While, I understand that is probably not the focus of the paper, without some justified rationale, it is difficult to translate these findings when any of the assumptions or the control variables change.

**Minor formatting/Grammar errors:**

Please note that following items did not affect my score. I understand errors tend to naturally creep up when preparing a manuscript and I am pointing that out merely to improve the quality of the draft.

- "TM" is incorrectly included in the citation in Section 2 (just before equation 6).
- Extraneous "'s" at the end of the first line of Section 3.2.
- Minor Typo in Section 3.2: "Figure 3 (b) and (c) show the **e**ffect of different heads ...". Same spelling error is present in other places in the manuscript.

**Questions:**

- Given that you have fixed the other parts of the TM (not related to the head module) fixed, would your findings change if any of those other components are chosen differently?

---

> ### Author Response · Authors · 2025-11-30
>
> **Q:** Given that you have fixed the other parts of the TM (not related to the head module) fixed, would your findings change if any of those other components are chosen differently?
>
> **A:** We fixed the backbone architecture to DiT as it already received much attention and exploration in past works and is proven effective as a diffusion/flow-matching model across various modalities [2]. Based on the current theoretical understanding of DTM, we do not anticipate significant changes in our findings if alternative backbone architectures were employed. Furthermore, we have conducted a few ablations over the backbone, such as time weighting (see Table 2a) and parametrization (see Figure 12). Nevertheless, we acknowledge that further investigation into different backbone choices could potentially provide additional insights.
>
> *[2] Scalable Diffusion Models with Transformers, Peebles and Xie, 2023.*\
> \
> \
> **Q:** Formatting/grammar errors.
>
> **A:** Thank you for pointing out the minor formatting and grammatical errors. We appreciate your attention to detail and will ensure that all such issues are corrected in the final draft. Regarding the use of “effect” we kept it as a noun where it makes sense.\
> \
> \
> **Q:** Lack of theoretical discussion.
>
> **A:**  (As we noted in our answer to Reviewer 4riu): While we agree on its importance, we want to highlight that the current theoretical understanding of generalization in flow and diffusion models is extremely limited and largely relies on empirical evidence (e.g., [1]). By choice, we decided to focus in this paper on providing the community with a comprehensive set of empirical data for the Transition Matching framework. We strongly believe that such data-driven studies are necessary to establish new paradigms (if not theoretical understanding) and advance the field.
>
>
> *[1] On the Closed-Form of Flow Matching: Generalization Does Not Arise from Target Stochasticity, Bertrand et al. 2025.*

---

### Official Review · Reviewer_4riu · 2025-11-03

**Soundness:** 3
**Presentation:** 4
**Contribution:** 3
**Rating:** 8
**Confidence:** 5

**Summary:**

This paper systematically investigates the design space of Transition Matching (TM) models for generative modeling, focusing on the time-continuous bidirectional variant and the architectural and algorithmic choices of the "head" module. The authors present a thorough empirical study spanning 56 different 1.7B parameter text-to-image models and 549 total evaluations, exploring factors such as head type (MLP, Convolution, Transformer), sequence scaling, batch size, time weighting, model parameterizations, and sampling strategies. The main findings are that TM with an MLP head, specific time weighting, and a high-frequency stochastic sampler gives top performance across broad metrics, while a Transformer head with sequence scaling rivals in aesthetic quality. The paper provides actionable recommendations for practitioners and positions TM as a state-of-the-art competitive approach among contemporary generative models.

**Strengths:**

- Comprehensiveness: The paper offers an admirably thorough experimental exploration, running significant computational experiments (56 trained large-scale models, 549 evaluations) that are rarely matched in scope in generative modeling work.
- Systematic Ablation: The design/practice space is cut along multiple axes—head type, head size, sequence scaling, batch, time weighting, parameterization, and samplers—granting nuanced insights into what factors matter for TM performance.
- Solid Empirical Support: Results are benchmarked across four diverse prompt datasets and 25 evaluation metrics, aggregated and analyzed to prevent cherry-picking, which is evident in the detailed presentation in Table 4 and the ablation tables/figures.
- Clear Positioning and Contributions: The roles of MLP vs. Transformer heads are well interrogated, and findings (such as sequence scaling’s effect in Transformer heads) are substantiated in both plots (see Figure 4) and tables, enabling actionable and evidence-based recommendations.
- Efficiency-Quality Tradeoffs: Generated data (see Figures 1, 7, 8, 9) visually and quantitatively demonstrates not only state-of-the-art quality but also improved generation cost and inference speed, which is particularly valuable to practitioners.
- Mathematical Clarity: Mathematical formulations (see Equations 3–13, especially the ODE-based head sampling) are overall sound, tying TM methodology to both flow/diffusion formalisms and exposing the degrees of freedom unique to TM (e.g., supervisory process, $Y$ parameterization).

**Weaknesses:**

- While the empirical exploration is outstanding, the theoretical explanation for why certain design changes—such as the specific benefit of token-wise MLP heads for text alignment, or the stochastic sampler’s effectiveness—remain largely empirical. There is limited grounding in theory or analysis for these effects, and at points, the paper admits uncertainty ("It is not clear to the authors..."), which limits the generalizability and explanatory power of reported findings.
- While the paper competently implements and evaluates strong baselines, the connection to recent alternative methods—including energy-based, equilibrium, and posterior mean matching approaches—is absent; no experimental or discussion-based comparison is made, which hinders the completeness of the empirical evaluation and weakens claims to "SOTA".

**Questions:**

Please refer to Weaknesses

---

> ### Author Response · Authors · 2025-11-30
>
> **Q:** Limited grounding in theory or analysis.
>
> **A:** We appreciate the emphasis on the need for stronger theoretical grounding. While we agree on its importance, we want to highlight that the current theoretical understanding of **generalization in flow and diffusion models** is extremely limited and largely relies on empirical evidence (e.g., [1]). By choice, we decided to focus in this paper on providing the community with a comprehensive set of empirical data for the Transition Matching framework. We strongly believe that such data-driven studies are necessary to establish new paradigms (if not theoretical understanding) and advance the field.
>
> *[1] On the Closed-Form of Flow Matching: Generalization Does Not Arise from Target Stochasticity, Bertrand et al. 2025.*\
> \
> \
> **Q:** Missing baselines such as energy-based, equilibrium, and posterior mean matching
>
> **A:** We appreciate your suggestion to include comparisons with recent alternative methods, such as energy-based, equilibrium, and posterior mean matching approaches. We note however that the mentioned baselines (energy-based, posterior mean matching, and equilibrium matching  which is in fact a concurrent ICLR submission) while potentially promising are still not established as SOTA at scale, and as far as we are aware, not scaled to the scale discussed in this paper (pretrain of 1.7B model with >300M images training set). Instead we focused on already widely adopted and well established baselines, including both discrete and continuous autoregressive/masked autoregressive models, as well as flow-matching models.

---

### Meta-Review · Area_Chair_QFyv · 2026-01-06

**Summary:**

The paper performs a large-scale exploration on the design space and training practices of Transition Matching, an emerging and unifying paradigm for generative modeling. Reviewer scores are somewhat divergent (8, 8, 4, 4, 6), primarily depending on difference perspectives on whether a systematic exploration constitutes sufficient technical novelty. In the rebuttal, the authors argue that progress in generative modeling has historically relied on such large-scale empirical investigations, particularly when theoretical understanding remains limited. AC agrees more on the authors’ perspective, and believes that the paper makes a meaningful contribution to the field that is worth presenting at ICLR. Accordingly, AC supports acceptance.

**Reviewer Concerns:**

Overall, the reviewers raised the following major concerns:

- Limited perceived technical novelty [gvxM, fSgD]
- Lack of theoretical grounding or explanatory analysis [4riu, fGYX, fSgD, yyoT]
- Limited experimental scope [gvxM, yyoT]

The rebuttal clarified that the paper aims to provide a systematic empirical foundation for the Transition Matching paradigm within a controlled scope, and that such large-scale exploration is particularly valuable given the limited theoretical understanding of the area.

**Reviewer Scores:**

- Reviewer 4riu: Initially 8. Would maintain the original positive score.
- Reviewer fGYX: Initially 8. Would maintain the original positive score.
- Reviewer gvxM: Initially 4. The score would likely increase to 6 or above.
- Reviewer fSgD: Initially 4. The score may remain unchanged upon the rebuttal, depending on the reviewer’s perspective.
- Reviewer yyoT: Initially 6. Would maintain the original positive score.

---

### Decision · Program_Chairs · 2026-01-26

Accept (Poster)